# Recent Research on *Cannabis sativa* L.: Phytochemistry, New Matrices, Cultivation Techniques, and Recent Updates on Its Brain-Related Effects (2018–2023)

**DOI:** 10.3390/molecules28083387

**Published:** 2023-04-12

**Authors:** Laura Siracusa, Giuseppe Ruberto, Luigia Cristino

**Affiliations:** 1Istituto di Chimica Biomolecolare, Consiglio Nazionale delle Ricerche, Via Paolo Gaifami, 18, 95126 Catania, CT, Italy; 2Istituto di Chimica Biomolecolare, Consiglio Nazionale delle Ricerche, Via Campi Flegrei, 34, 80078 Pozzuoli, NA, Italy

**Keywords:** *Cannabis sativa* L., secondary metabolism, cannabinoids, minor compounds, alternative *Cannabis* matrices, controlled growth, plant and soil microbiota, neurodevelopmental and neurodegenerative effects

## Abstract

*Cannabis sativa* L. is a plant that humankind has been using for millennia. The basis of its widespread utilization is its adaptability to so many different climatic conditions, with easy cultivability in numerous diverse environments. Because of its variegate phytochemistry, *C. sativa* has been used in many sectors, although the discovery of the presence in the plant of several psychotropic substances (e.g., Δ^9^-tetrahydrocannabinol, THC) caused a drastic reduction of its cultivation and use together with its official ban from pharmacopeias. Fortunately, the discovery of *Cannabis* varieties with low content of THC as well as the biotechnological development of new clones rich in many phytochemical components endorsed with peculiar and many important bioactivities has demanded the reassessment of these species, the study and use of which are currently experiencing new and important developments. In this review we focus our attention on the phytochemistry, new matrices, suitable agronomic techniques, and new biological activities developed in the five last years.

## 1. Introduction

The origins of *Cannabis sativa* in terms of its cultivation and use date back about 12,000 years, in the regions of Central and East Asia. Its diffusion around the world has been continuous and inexorable [1], due to the easy adaptability of this plant to very different climate conditions, and thanks also to its numerous properties and uses as a fiber, food, and drug [2,3,4,5,6].

*C. sativa* is an herbaceous and dioecious plant belonging to the order Urticales and the Cannabaceae family [5]. The taxonomic description of the *Cannabis* genus is almost controversial; ‘sativa’ is considered de facto the sole species of the genus. Even though other species were suggested, ‘*indica*’ and ‘*ruderalis*’ being the most important, these different species are today considered merely varieties. 

Concerning its content of secondary metabolites, *C. sativa* has been well studied and as of today about 750 compounds have been reported in this plant [5,7]. 

Cannabis has been widely cultivated due to its industrial [8], ornamental [9], nutritional [10], medicinal, and recreational potential [11]. From regulatory and application perspectives, cannabis plants are categorized based on the level of Δ^9^-tetrahydrocannabinol (THC), one of the most important phytocannabinoids [12]. Plants are generally classified and regulated as industrial hemp if they contain less than 0.3% THC in the dried flower (this level varies by country) or as a drug type with more than this threshold [13]. Furthermore, the study of the phytochemical profile of *C. sativa* has recently been demonstrated useful not only for determining the potential toxicity of *C. sativa*, considering the widespread use of this plant for human consumption and as animal feed, but also for traceability purposes [14,15,16]. The aim of this review is to provide an update of the studies carried out on *C. sativa* in the past five years, dealing with different aspects such as phytochemistry, novel cultivation techniques, and newly described biological activities especially connected with the effects of cannabis on the central nervous system.

## 2. Secondary Metabolites of Cannabis

### 2.1. Cannabinoids

*C. sativa* contains a peculiar class of phytochemicals known as phytocannabinoids, present almost exclusively in its inflorescences. These substances have also been identified in rhododendron, licorice, and liverwort. From a chemical point of view, cannabinoids are meroterpenoid compounds with a basic structure of a resorcinyl moiety with different isoprenyl, alkyl, or aralkyl side chains. Tetrahydrocannabinol (THC) and the corresponding acid tetrahydrocannabinolic acid A (THCA-A), cannabidiol (CBD), cannabinol (CBN), and cannabidiolic acid (CBDA) are the main components of this class; altogether, over 200 different cannabinoids have been isolated from the leaves of *C. sativa* [2,7,17,18,19]. A selection of the main cannabinoids found in *C. sativa* is listed in Table 1 and Figure 1.

These compounds have attracted the interest of humans including researchers, owing to the broad possibility of their exploitation in different sectors and also for their biological activities. In particular, interest in these molecules has exponentially increased with the discovery in the human body of the endocannabinoid system and the cannabinoid receptors: CB1 and CB2 [20,21]. The study of the endocannabinoid system allowed better understanding of the medicinal effectiveness of cannabis in the treatment of different pathologies. However, the discovery of the psychotropic effects associated with Δ^9^-THC led to the ban of *C. sativa* from the USA pharmacopoeia and subsequently in many other countries.

On the basis of the THCA/CBDA ratio, several chemotypes (Table 2) have been individuated: chemotype I (plants with a THCA/CBDA ratio > 1.0), chemotype III (THCA/CBDA < 1.0 or fiber plants) and chemotype II (intermediate THCA/CBDA ratio). Finally, plants with a very high content of cannabigerolic acid (CBGA) are classified as chemotype IV, and plants with no cannabinoids belong to chemotype V [22]. It is important to underline that the term “chemotype” refers to plants of the same genus that are totally identical in terms of botanical and taxonomic features and differ only in the profile of the specialized metabolites that they are able to synthesize and accumulate [2,23]. 

Today, *C. sativa* is classified into two main types depending on the amount of Δ^9^-THC: the drug type with Δ^9^-THC > 0.3% is known as cannabis (marijuana), and the fiber type with Δ^9^-THC < 0.3% is commonly referred to as hemp. 

THCA is the main cannabinoid in the drug-type plant, while CBDA is the main cannabinoid in the fiber-type hemp [24]. Both acids undergo decarboxylation during storage and in general with heating, giving rise to the phenols THC and CBD, respectively [24]. While the use and cultivation of drug-type cannabis is subject to legal restrictions in many countries owing to the psychotropic effect of Δ^9^-THC, the hemp type is legal and freely cultivated [4,25].

Notwithstanding the large number of cannabinoids that have been isolated and characterized from *C. sativa*, its phytochemical studies keep provoking interest to the point that new cannabinoids are being continuously isolated and characterized (Figure 2). 

The recently discovered cannabinoid Δ^9^-Tetrahydrocannabiphorol (Δ^9^-THCP)was initially known as synthetic derivative, but has been successfully isolated from *C. sativa*, resulting in a product 30 times more active than Δ^9^-THC, the main psychotropic compound of marijuana [26]. More recently, the racemic couple of *cis*-stereoisomers of Δ^9^-THC were isolated from a low-THC-containing variety of *C. sativa* registered in Europe as fiber hemp. Although the in vivo assay of psychotropic activity revealed that they are mostly inactive in mice, further analyses are necessary to confirm this aspect [27]. Another newly discovered cannabinoid has been called cannabitriol and can be considered a new CBD derivative [19]. Further cannabinoids have recently been characterized, such as cannabiwinol CBDD, anhydrocannabimovone, isolated for the first time as a natural product, and three new hydroxylated CBD analogues: 1,2-dihydroxycannabidiol, 3,4-dehydro-1,2-dihydroxycannabidiol, and hexocannabitriol, obtained from the crystallization of CBD (Figure 2) [28]. 

Phytocannabinoids represent the most important and promising specialized/secondary metabolites from *C. sativa* L. The peculiar characteristics of these compounds, as well as their high levels of diverse bioactivities have allowed the production and approval of several cannabis-derived drugs. Despite over 150 cannabinoids already having been identified, research into new cannabinoids is particularly lively in the aspects of synthetic chemistry and biotechnology. However, some challenging problems remain, including the clarification of the complex biosynthetic route of cannabinoids as well as the optimization of the biotechnological processes [11].

### 2.2. Terpenoids

Terpenoids represent another important class of secondary metabolites of *C. sativa*. From a biosynthetic point of view, the previously described cannabinoids can also be defined as meroterpenoids. To date, about 200 terpenoids have been detected in *C. sativa*, mainly in leaves, roots, and flowers, as well as in the form of essential oils [4,8,13,15,19,29,30,31]. Most of the terpenes isolated from *C. sativa* come from the hydrodistillation of the inflorescences. A selection of volatile components obtained by hydrodistillation is reported in Table 3.

It is interesting to point out that most of these essential oils contain considerable amounts of cannabinoids, the molecular weight and volatility of which are reflected in the essential oil from *C. sativa* [4,17,24]. 

### 2.3. Polyphenols

Polyphenols represent a class of specialized metabolites almost ubiquitous within the plant kingdom. Although flavones and flavonols (apigenin, luteolin, kaempferol, and quercetin) are well represented in *C. sativa* aerial parts, this species also contains unique flavonoids such as cannaflavins A, B, and C, all methylated isoprenoid flavones [25] (Figure 3). Flavonoids constitute about the 10% of the compounds in *C. sativa*, and to date about 26 flavonoids have been recognized [32,33]. Stilbenoids are a further class of phenolic components found in hemp. So far, nineteen compounds belonging to this class have been isolated from *C. sativa*: canniprene, a unique dihydrostilbene, and three new dihydrostilbenoids are reported in Figure 3. These compounds have all been isolated from *C. sativa* leaves together with known stilbenoids [4,34].

Among the polyphenolic metabolites found in *C. sativa*, a group of peculiar phenolic amides named phenylproprionamides, also known as cannabisins, are worthy of mention (Figure 4). Until now, about 20 cannabisins have been isolated from hempseed. The exact origin of these compounds is not totally clear, most probably they are produced by a coupling reaction between hydroxycinnamic acid amids (HCAAs) and rare phenolic amides [35,36,37,38]. Biological tests carried out on these particular compounds have revealed their potential as antioxidant and acetylcholinesterase inhibitors, suggesting their suitability in the treatment of Alzheimer’s disease [39].

### 2.4. Alkaloids

Alkaloids are another important class of secondary metabolites identified in *C. sativa* and isolated from its leaves, stems, pollen, roots, and seeds. Cannabisativine, anydrocannabisativine, and the indole alkaloid dihydroxyisoechilunilina (Figure 5) are representative examples [4,40].

## 3. Alternative Cannabis Matrices

### 3.1. Cannabis Roots

As mentioned above, *C. sativa* is a plant with a very ancient tradition. It can be used to obtain a variety of products useful in many sectors of human daily life. In the previous chapter we have shown, in a non-exhaustive selection, some of its most interesting phytochemical components. However, to obtain a more complete picture of this plant it is necessary to take into account other parts of the plant that have received scarce interest, such as roots, seeds, and seed hulls. 

In the phytotherapeutic sector, the interest in this plant has been devoted to its inflorescence and leaves, where the most important bioactive components cannabidiol (CBD) and Δ^9^-tetrahydrocannabinol (THC) are mainly concentrated. The roots have been comparatively neglected, and have not received the same attention as the aerial parts, although canapa roots have been used to fight fever, inflammation, infections, and arthritis [41,42]. Phytochemical analyses of roots did not show the presence of cannabinoids, or only in traces, whilst the most important components of this matrix resulted in a series of triterpenoids subdivided between triterpenes and phytosterols, some of which are reported in Figure 6 [1]. The presence of triterpenoidic compounds in the roots of *C. sativa* was recently confirmed by Ferrini et al. [43]. They suggest the application of aeroponic cultivation, which would facilitate all the relevant agronomic processes by being easy, standardizable, and favoring the growth of the plant’s root system compared with conventional soil-based cultivation methods. 

A further and more recent phytochemical study on canapa roots was carried out by Oh et al. [44], who worked with a sample of hemp cultivated in Korea. Briefly, after water washing and air drying, the roots were analyzed by HPLC-DAD, which allowed researchers to establish the presence and the amounts of *p*-coumaric acid and ethyl-*p*-coumarate as the main constituents of the root extracts (Figure 7). According to their findings, the authors suggested the industrial exploitation of the hemp roots of *C. sativa* as a source of p-coumaric acid derivatives, in consideration of their broad biological activities.

A very recent study was carried out by an Argentinian research group, aiming to determine the antibacterial activities of extracts from inflorescences and roots of *C. sativa* [45]. The content of total phenolic compounds, total flavonoids, and total saponins was determined by applying previous methodologies [46,47,48,49].

Antibacterial activity against *Paenibacillus larvae*, *Staphilococcus aureus*, and *Esherichia coli* was measured by the MIC methodology; the recorded activities even if not very significant were associated with the high content of triterpenoids and saponins.

Many research groups have reported cannabinoids to be absent or present only at trace levels in the roots of *C. sativa*. However, this suggestion has recently been challenged [49]. Furthermore, a Brazilian research group recently reported the presence of a new cannabinoid called Δ^9^-tetrahydrocannabutol (known also as tetrahydrocannabinol-C4—THC-C4), that is structurally very similar to the main cannabinoid Δ^9^-THC, the sole difference being a butyl side chain instead of a pentyl side chain. Further, four nitrogenous compounds were characterized and all are reported in Figure 8 [50].

### 3.2. Cannabis Seeds and Seedhulls

Three different types of oil can be obtained from *C. sativa*: 

(a) vegetable oil, obtained by extraction of seeds and constituted mainly by triglycerides, suitable for human consumption owing to a good balance of ω-6 and ω-3 essential fatty acids (EFAs); 

(b) essential oil from inflorescences and flowers, obtained by hydrodistillation and steam distillation; 

(c) so-called ‘hashish oil’, normally obtained by solvent extraction from plants [1].

With the aim to exploit each fraction from the workout of hemp *Cannabis sativa*, seed hulls from *C. sativa* have been also investigated for their content. Using chloroform as extracting solvent and HPLC/MS as an analytical tool, nine diacylglycerols (DAGs), six lysophatidylcholine (LPCs), five lysophosphatidylethanolamines (LPEs), eight phosphatidylethanolamine (Pes), and thirteen phosphatidylethanolamines (PCs) were recovered from the matrix [51]. These new compositional data add another input to the list of uses of this ancient plant.

Fibers represent a considerable portion (>77%) of *C. sativa* L. and therefore the main component usable for further industrial processes. Recently, an Australian group demonstrated a large increase in fiber solubility by applying “extrusion technology”; the authors also observed enhancement of the binding capacity of water and oil, delayed starch gelatinization, and inhibition of retro-degradation. The supplementation of this material into animal feed showed a positive effect on animal microbiota [52].

An international research team carried out a study on eight varieties of whole hemp seeds, and on eight commercial samples of dehulled hemp seeds, with the aim of determining the seeds’ chemical composition and biological properties [53]. The compounds analyzed were soluble sugar and phenolic compounds, and the tested biological properties included antioxidant, cytotoxic, and antimicrobial activity. Sugars were the main components in both samples, and whole and dehulled seeds showed similar soluble sugar profiles, containing fructose, glucose, sucrose, and raffinose. Regarding their phenolic compounds, the two studied matrices in this case showed similar profiles, with the ferulic acid hexoside and syringic acid as the main components. The biological activities were interesting and the authors concluded that hemp seed has optimal potential as a functional food.

Considering that edible seeds are today largely present in human and animal diets an Indian group has recently focused its attention on the nutritional and pro-health properties of the five most popular edible seeds: Chia (*Salvia hispanica* L.), Hemp (*Cannabis sativa* L.), Pumpkin (*Cucurbita*), Sunflower (*Helianthus annus*) and Safflower (*Carthamus tinctoria*) [54,55]. Concerning hemp the authors report that 100 g of seeds contain 553 kcal of energy and possesses digestible proteins, polyunsaturated fatty acids (PUFA), lipids and carbohydrate. The omega-6 compounds in the total content of fatty acids ranges from 64 to 72%. Oil from *C. sativa* seeds shows traces of cannabidiol, β-sitosterol, and terpenes together with the tocopherol molecular group including α-, γ-, β-, and δ-tocopherol. The presence of the three main nutrients, namely healthy fat, protein, and fiber provide to the seeds a big opportunity of success, which is also supported by a significant diffusion of vegan/vegetarian diets among the people. Always under the nutritional point of view of *C. sativa* seed oil it is important to underline that the seeds are particularly rich in omega-3 and stearidonic acids [56].

### 3.3. Cannabis Leaves as Functional Foods

Roots, stems, and leaves from *C. sativa* have recently been investigated to determine their content of tryptophan (TRP) and its metabolic products: kyurenine (KYN) and kynurenic acid (KYNA), which have biological properties (Figure 9). Since these compounds are present in higher quantities in the leaves of *C. sativa*, this matrix is proposed as alternative source for these substances in the field of functional foods [57]. 

## 4. Cultivation Techniques Especially Effective on *Cannabis sativa*—Recent Updates

### 4.1. The State of the Art

As previously stated, secondary metabolites in *C. sativa* (mainly cannabinoids, terpenoids, and polyphenols) are produced in all aerial parts of the plants, particularly in the leaves and female inflorescences. Their production, such as that of all species belonging to the vegetable kingdom, is related to genetic, environmental, and agronomic factors (plant chemotype, growth conditions, and phenological stage at harvest) [58]. This high variability of growth conditions inevitably leads to broad variability in the content of the plant’s secondary metabolites, which represents the main problem for the use of cannabis for medical purposes. To overcome this intrinsic problem, several strategies have been developed, including asexual propagation of selected varieties and highly technological indoor cultivation [59,60,61]. Asexual reproduction of cannabis requires only a single plant, allowing the multiplication of a single genotype for commercial production of individuals that meet the desirable pharmaceutical traits [62]. Since indoor cultivation in the controlled conditions of high-technology greenhouses carries significantly high manufacturing costs, attention has been directed towards biomass production and the content of bioactive substances in order to compensate for the expense [63]. An alternative approach is to improve and control the secondary metabolism of plants in open fields through the use of selected new-generation fertilizers or by controlling plant and soil microbiota. These strategies are discussed in detail in this section. It is worth mentioning that agronomic techniques can be replaced (or associated) with molecular approaches such as the sequencing of the cannabis genome, and the manipulation of the cannabis plant itself for the altered production of specific compounds, in order to produce medically relevant cannabis varieties with elevated concentrations of specific molecules [64].

### 4.2. Soilless Growing Technologies Applied to Cannabis: Hydroponic, Aquaponic, and Aeroponic Cultivation—Recent Updates

Because of the aforementioned issues associated with the high intrinsic variability in *C. sativa* secondary metabolism, the majority of medical research involving the cultivation of this species occurs in controlled environments (hi-tech greenhouses and indoor facilities) making use of soilless cultivation systems such as hydroponics, aquaponics, and aeroponics. Hydroponics systems are well-known, in which plant roots are partially and/or totally immersed in a nutrient solution. Irrigation methods in hydroponics include drip irrigation (a nutrient solution is fed into an inert growing medium used merely as physical support for the root system), deep water culture (the plant root system is completely submerged in the nutrient solution and the plants are supported by a membrane system that blocks the immersion of the aerial parts), nutrient film techniques (only the lower portion of the root system is immersed in a flowing nutrient solution, whilst the upper roots are exposed to the air), and flood and drain (plant roots are immersed in a nutrient solution for a period of time, subsequently drained and collected in a reservoir to aerate the root bed) [65]. Aquaponics is a soilless cultivation system that has recently gained scientific and commercial interest due to its circular approach and reduced environmental impact [66]. Aquaponics uses fish effluent and naturally occurring bacteria to fertilize plants in a circular ecosystem that produces fish and plant materials simultaneously. With this approach, the utilization of fertilizers and pesticides is highly reduced, providing an ecofriendly system. The comparable yields obtained from aquaponic cultivation when compared with conventional systems can be attributed to a plethora of diverse plant-growth-promoting microorganisms that can improve efficiency of nutrient use.

The third soilless approach useful for indoor *C. sativa* cultivation is aeroponics, in which the plant is entirely exposed to the air and the nutrient mixtures are nebulized to plant root systems in the form of droplets, typically of 10–100 µm diameter. The most commonly used aerosol generation technology in aeroponics is high-pressure atomization, where high-pressure liquids are forced through a small orifice, breaking the liquid stream into droplets [67]. 

### 4.3. LED Lighting Techniques

In the so-called “ecological pyramid”, plants are classified as sole producer organisms, occupying only the pyramid basement. Members of the vegetable kingdom are able to synthetize glucose, the molecule at the center of all metabolic charts, starting with carbon dioxide and water and using solar light as a catalyst. It is therefore self-evident that a close relationship exists between plants and light: plant morphology, metabolism, growth, and development are strongly influenced by circadian cycles and can be artificially manipulated by the quality and duration of light [68,69,70]. In the indoor cultivation scenario, the use of artificial lights for plant cultivation, particularly that of light-emitting diode (LED) technology, can result in significant reductions in energy consumption and in the achievement of desirable yields and phytochemical traits. In fact, it has been demonstrated that different light wavelengths have effects on different physiological features of the plant such as pigmentation, secondary metabolite production, chloroplast development, photosynthetic activity, and plant biomass accumulation, to cite just a few [71,72]. In *C. sativa*, due to the previously described complexity of its secondary metabolism, light quality and quantity have significant effects on the biologically active components. For example, UV-B radiation (280–315 nm) did not impact cannabinoid content in cannabis plants with the exception of Δ^9^-THC, whilst UV-A radiation (315–380 nm) from full-spectrum LED arrays induced an increase of several compounds including CBD, CBG, Δ^9^-THC, and tetrahydrocannabivarin (THCV) [73]. Recently, Moher et al demonstrated that not only the choice of light radiation but also its intensity can induce modifications in *C. sativa* plants’ morphology and growth processes [74]. The authors did not mention the effects of the parameters they applied on the secondary metabolic profiles of cannabis plants, which can surely represent a further step of knowledge in this field. 

### 4.4. Cannabis Symbiotic Microorganisms: A Yet Unexplored Scenario 

In the past decade, the study of microbial communities associated with higher living organisms, known as “beneficial microbes”, has gained exponential interest due to its significant effects on human health. The composition and content of microbial species hosted in several parts of the human body, particularly those residing in the gut, have been intensively studied by a plethora of multidisciplinary approaches. In spite of innumerable works in the literature dealing with human microbiota, those focused on the microbiota of agricultural plants are relatively few. Indeed, this is due to the high number of plant species involved and the fact that plant microbial communities are heavily affected by a number of factors including developmental stages, root exudation, soil type, and environmental conditions. Although not the focus of the present work, approaches involving the manipulation of *C. sativa* microbiota are definitely worthy of mention. For an extensive review of this specific topic, see the recent paper by Taghinasab and Jabaji [75].

Owing to its rich and variegate secondary metabolic pool, *C. sativa* constitutes a perfect model to explore plant–microbiome interactions and to assess how endophytes can modulate the production of secondary metabolites. Endophytes have gained a prominent role in the list of abiotic and biotic elicitors for the growth, mass yield, and metabolic performance of this plant. Pagnani et al. reported that a mixture of four bacterial species applied to the roots of cannabis seedlings in vitro efficiently colonized the entire root system of the plant, favoring plant growth and development in greenhouses as well as accumulation of secondary metabolites. Particularly, two of the main cannabinoids from cannabis, CBD and THC, were significantly improved by endophytic treatment [76]. Nowadays, even though basic information on the diversity and composition of cannabis endophytes has been published, the majority of the works deal with the isolation and identification of microbes rather than their modulating effects on the plant’s secondary metabolic pool. Therefore, systematic study of this topic is desirable.

## 5. Brain-Related Effects of Cannabis Use: Risks and Beneficial Impact in Users and Patients 

The legalization of cannabis in many countries, as well as the decrease in perceived risks by the users, have contributed to the increase in recreational consumption. However, despite the widespread use and misuse of cannabis, little remains known about its putative harmful cognitive and neurobiological effects, possibly associated with its psychoactive risks as a “drug of abuse” [77,78,79]. To disentangle divergent results about risk factors for users from therapeutic benefits for patients, this section summarizes studies searched in MEDLINE and published in English between 2018 and March 2023. According to current knowledge, no formal study has shown cognitive impairment in medical cannabis patients. Meta-analysis of data from non-medical cannabis users among teenagers and young adults reveals non-statistical salience after 72 h of abstinence and no evident permanent sequelae [80]. From the biological point of view, ∆^9^-tetrahydrocannabinol (THC) is the major euphoric component of *C. sativa* [81,82] followed by minor euphoric cannabinoids such as cannabinol (CBN) and cannabinodiol, while cannabidiol, cannabigerol (CBG), cannabichromene (CBC), cannabidivarin (CBDV), and ∆^9^-tetrahydrocannabivarin (THCV) are non-euphoric compounds. Today, over a hundred cannabinoids have been isolated from *C. sativa*. The term ‘cannabinoids’ refers to the oxygen-containing C21 aromatic hydrocarbon compounds naturally occurring in the cannabis plant. However, the term “phytocannabinoids” is also used to distinguish them from the synthetic compounds that mimic their chemical structures and functions rather than their botanical ones [83]. Although a large part of the dichotomy between the risk and benefits of cannabis is attributable to variability in the cannabinoid composition within the plant itself, only THC administration (and to a lesser degree cannabinol administration) produces similar effects in mice to those of marijuana in humans, such as catalepsy, analgesia, hypolocomotion, and hypothermia [84]. Consequently, THC is considered the major psychotropic component of recreational cannabis preparations. In contrast, there is no evidence that THC mediates the medicinal effects of cannabis, whereas CBD is recognized as the most relevant cannabinoid with therapeutic potential in several disorders.

The empirical use of the cannabis plant has been practiced for millennia in many cultures to alleviate pain, migraine, and inflammation. The medical and therapeutic application of cannabis began to be well understood from the 1960s, when the structure and functional expression of the first G protein-coupled receptor of cannabinoids, named CB1, were characterized in brain tissue [85,86,87], followed by the GPCR cannabinoid receptor 2 (CB2) which was later identified by homologous cloning in the cells of the immune system [88]. In turn, these discoveries led to identification of endogenous CB1 and CB2 ligands, the lipids 2-arachidonoylglycerol (2-AG) and anandamide (the ethanolamide of arachidonic acid) named “endocannabinoids” [89,90]. The enzymes involved in their biosynthesis were later identified as N-acylphosphatidylethanolamine (NAPE)-specific phospholipase D-like hydrolase (NAPE-PLD), catalysing the synthesis of anandamide, and other *N*-acylethanolamines, along with diacylglycerol lipase α (DAGLα) and DAGLβ, which catalyze the biosynthesis of 2-AG and other monoacylglycerols. The enzymes involved in endocannabinoid inactivation were identified as fatty acid amide hydrolase (FAAH), catalyzing the hydrolysis of anandamide (and other *N*-acylethanolamines and fatty acid primary amides), and monoacylglycerol lipase (MAGL) which catalyzes the hydrolysis of 2-AG (and that of other monoacylglycerols) [91]. Altogether, the complex of cannabinoid endogenous signals, receptors, and metabolic enzymes is known as the “endocannabinoid system”. This system is complicated by its overlap with other pathways and alternative metabolic processes, merging into an expanded endocannabinoid system named “endocannabinoidome”. Starting from the common cannabinoid signal pathway through CB1 and CB2, which are expressed in brain regions involved in the neural processing of reward, habit formation, and cognition, the physiological effects of cannabinoids in the nervous system are exerted through various receptors, including adrenergic receptors, peroxisome proliferator-activated receptor-gamma, 5-HT_1A_, and the recently discovered GPCRs (GPR_55_, GPR_3_, and GPR_5_) [92]. The medical use of CBD has demonstrated safety, efficacy, and consistency sufficient for regulatory approval for treatment of spasticity in multiple sclerosis (MS) and in Dravet and Lennox–Gastaut syndromes (LGS). For other pathologies, the therapy approach is in phase II–III or under observational study (see Table 4). 

Further promising therapeutic approaches arise from recent discoveries regarding the efficacy of cannabis-based preparations against intractable epilepsy, brain tumors, Parkinson’s disease (PD), Alzheimer’s disease (AD), and traumatic brain injury (TBI)/chronic traumatic encephalopathy (CTE). Indeed, extensive observations of seizures in Dravet and Lennox–Gastaut syndrome (LGS) patients indicate good therapeutic success with lower doses of CBD than in Epidiolex^®^ when utilized in cannabis-based preparations with small concomitant amounts of THC, THCA, and linalool, a terpenoid component of cannabis [70,93,94]. In this regard, selective cannabis breeding via Mendelian techniques raises new therapeutical possibilities of producing chemovars with multiple anticonvulsant components that may offer synergistic benefits [95]. Phytocannabinoids offer cytotoxic benefits with effects on brain tumors that have been considered in relation to synergistic benefits of combinations of THC, CBD, and standard chemotherapy with temozolomide for glioma [96]. More recently, cannabis preparations with equal THC and CBD content combining THCA and even CBDA along with cytotoxic terpenoids such as limonene have been considered potentially useful in cancer treatment [72]. For AD patients, an extract of a type II chemovar of cannabis (THC/CBD) with a sufficient pinene fraction would seem to be an excellent candidate for clinical trials, providing multiple targeted benefits against symptoms such as agitation (THC, CBD, linalool), anxiety (CBD, THC—low dose, linalool), psychosis (CBD), insomnia/restlessness (THC, linalool), anorexia (THC), aggression (THC, CBD, linalool), depression (THC, limonene, CBD), pain (THC, CBD), memory (α-pinene, THC), and neuroprotection (CBD, THC). Finally, chemovars of cannabis combining THC and CBD, have been extremely helpful in treatment of dementia pugilistica or “punch-drunk syndrome”, traumatic brain injury (TBI), and chronic traumatic encephalopathy (CTE) by alleviating symptoms such as headache, nausea, insomnia, dizziness, agitation, substance abuse, and psychotic symptoms [97,98]. 

In recent times, special attention has been reserved for the use of cannabis by pregnant women, young mothers, and teens. Since the endogenous cannabinoid system plays an important role in neurodevelopmental processes in the prenatal/perinatal and adolescent periods, the consequence of cannabis abuse may be mainly attributable to epigenetic reprogramming that could maintain long-term impact into adulthood and across generations [99]. These negative outcomes can be amplified by individual genetic predispositions ultimately precipitating or worsening mental disorder [100]. Neuroimaging studies have shown that high doses of CBD induce significant alterations in brain activity and connectivity patterns in a resting state and during the performance of cognitive tasks, in both healthy volunteers and patients with a psychiatric disorder [78]. Larger studies are essential to reconcile divergent results for assessing potential moderators of cannabis effects on the developing brain; preliminary neuroimaging evidence reveals functional and structural alterations in frontoparietal, frontolimbic, frontostriatal, and cerebellar regions among adolescent cannabis users [101]. In recent times, each year almost 200 million people worldwide use cannabis for recreational or therapeutic purposes. However, the potential neuropsychiatric effects still need to be better understand, especially concerning possible induction of psychosis by cannabis consumption. In this regard, meta-analysis of data reviewed by Hindley et al. [102] examined the effect of THC, alone and in combination with cannabidiol (CBD), compared with placebo, on psychiatric symptoms in healthy people. They found that a single THC administration induced psychotic and other negative psychiatric symptoms, while no evidence indicated that CBD induces symptoms or counteracts the effects of THC. Furthermore, cross-sectional studies of adolescents and young adults report an association between cannabis use and cognitive dysfunction that may reflect residual effects from acute cannabis use or withdrawal [71].

## 6. Conclusions

In the time period considered by this review (2018–2023) a considerable interest in *C. sativa* has grown exponentially, as documented by the more than 3000 articles published on this species just in 2021. 

The topics covered in this work – phytochemistry, cultivation techniques and brain-related effects of *cannabis*—are only apparently unrelated to each other, since a leading thread exists and it is the highly variegated secondary metabolism of this plant that gives rise to a not so usual chemodiversity. In particular, the phytochemical studies carried out on cannabis allowed for establishing the toxicity aspects normally connected to an abuse of this plant, and at the same time how, for example, CBD is recognized as the most relevant cannabinoid with therapeutic potential in several disorders.

Since the phytocannabinoid composition is extremely inhomogeneous, and the ratios of active molecules are affected by a range of variables, the use of raw cannabis as a feedstock to reach the high-quality demands of the pharmaceutical industry is extremely challenging. For instance, to produce complex botanical medicine like Sativex^®^, which contains a specific THC and CBD ratio and a range of other molecules, GW Pharmaceuticals had to manage agricultural cannabis biomass selection, cultivation, and harvest/processing to ensure quality supplies for medical research to meet the prescription medicines uses for the treatment of spasticity due to multiple sclerosis. Similarly, in recent years, the production of CBD-based medicinal materials for research as potential therapeutics in childhood affected by epilepsy syndromes has come into greater attention for agricultural biomass selection and cultivation, to ensure quality supplies for medical research. To meet these requirements, an advanced Mendelian Cannabis breeding program has been exploited by pharmaceutical companies utilizing chemical markers to maximize the yield of phytocannabinoids and terpenoids with the aim to improve therapeutic efficacy and safety to produce selective chemovars with a therapeutical application for analgesic, anti-inflammatory, anticonvulsant, antidepressant, and anti-anxiety effects, while simultaneously reducing sequelae of Δ^9^-tetrahydrocannabinol such as panic, toxic psychosis, memory impairment. Integrated omics studies combining genomic data with metabolite profiles are now beginning to unravel the association between the expression of cannabinoid genes with THC:CBD ratio and cannabinoid content. Advanced agro-biotechnology methods could be further extended for recombinant production of cannabinoids in metabolically engineered hosts such as yeasts or bacteria. Pharmacological research coupled with rapidly evolving genome-based biotechnology applied to agriculture will further facilitate exploring cannabis plants for their remarkable potential in drug discovery. 

The interest in the phytochemical composition of cannabis is not only due to the possibility to use many of its phytochemicals in the medical sectors, but also to the industrial interest towards the same products. Many industrial applications of *cannabis* could not exist without the development of suitable and high-performance cultivation techniques. However, notwithstanding the large number of studies carried out in the phytochemical sectors, an exhaustive and complete characterization of cannabis extracts has not yet been reached, therefore new extraction procedures, as well as novel and more sophisticated characterization methodologies still have to be applied. 

In any case, cannabis phytochemistry, as well as agronomical and biotechnological techniques applied to its cultivation have benefited from this growing interest and from the trend observed in these years appears that the discovery of new specialized metabolites seems inevitable. In particular, as pointed out in this review, in the medical sector many new discoveries on the role of the metabolism of cannabis and how it interacts with the human central nervous system appear close to their discovery.

## Figures and Tables

**Figure 1 molecules-28-03387-f001:**
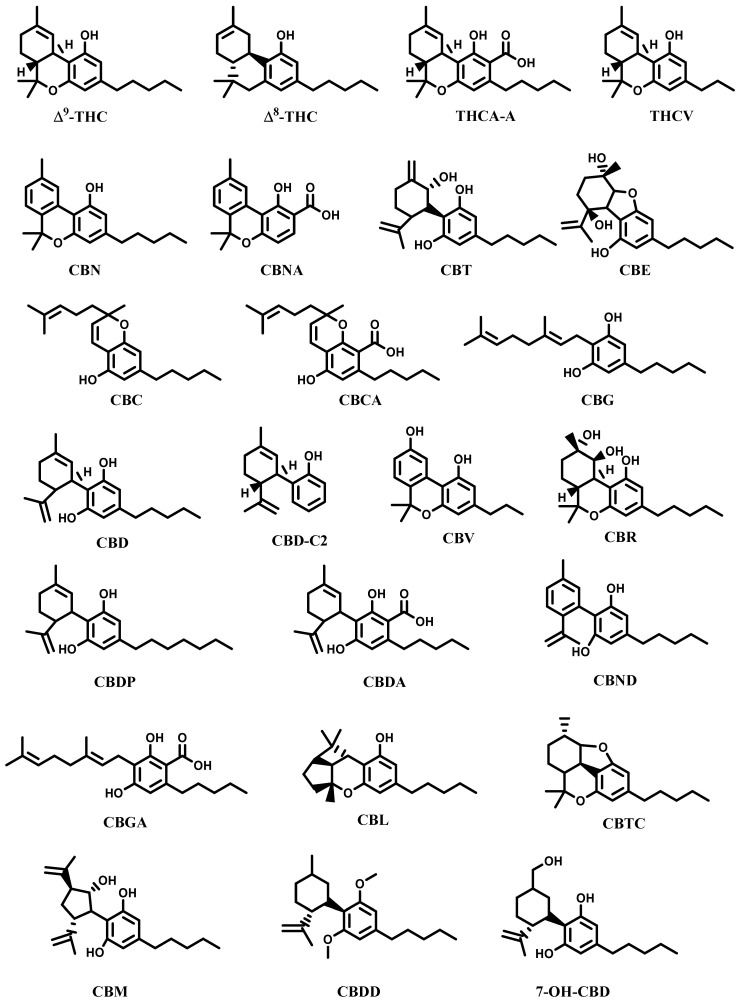
Selection of the main cannabinoids from *Cannabis sativa*.

**Figure 2 molecules-28-03387-f002:**
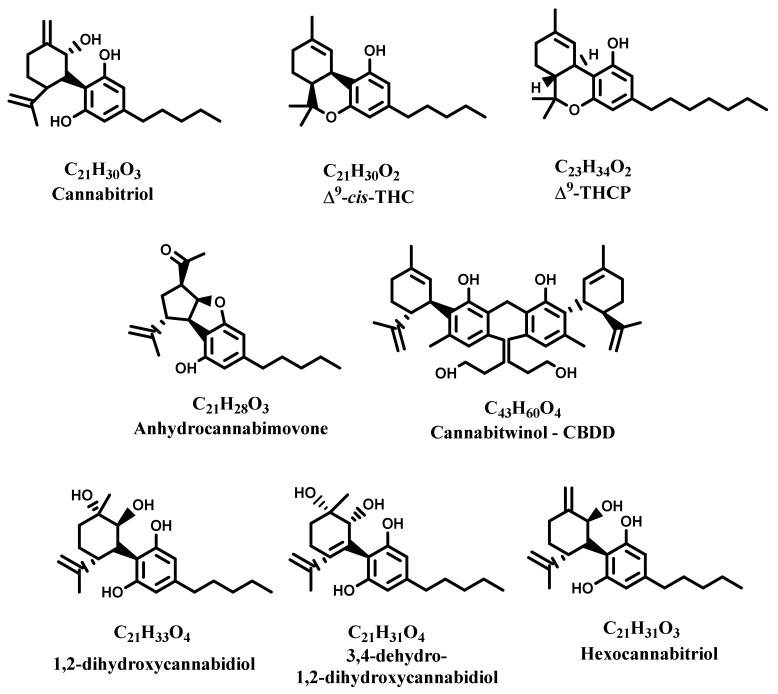
Recently isolated new cannabinoids.

**Figure 3 molecules-28-03387-f003:**
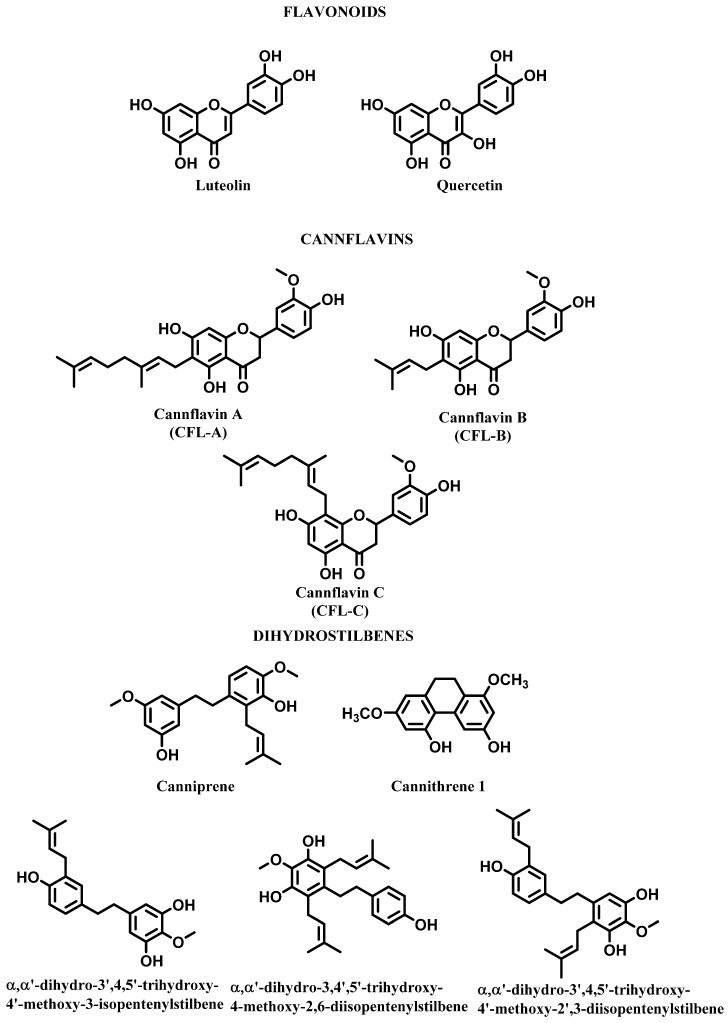
Polyphenols in *Cannabis sativa*.

**Figure 4 molecules-28-03387-f004:**
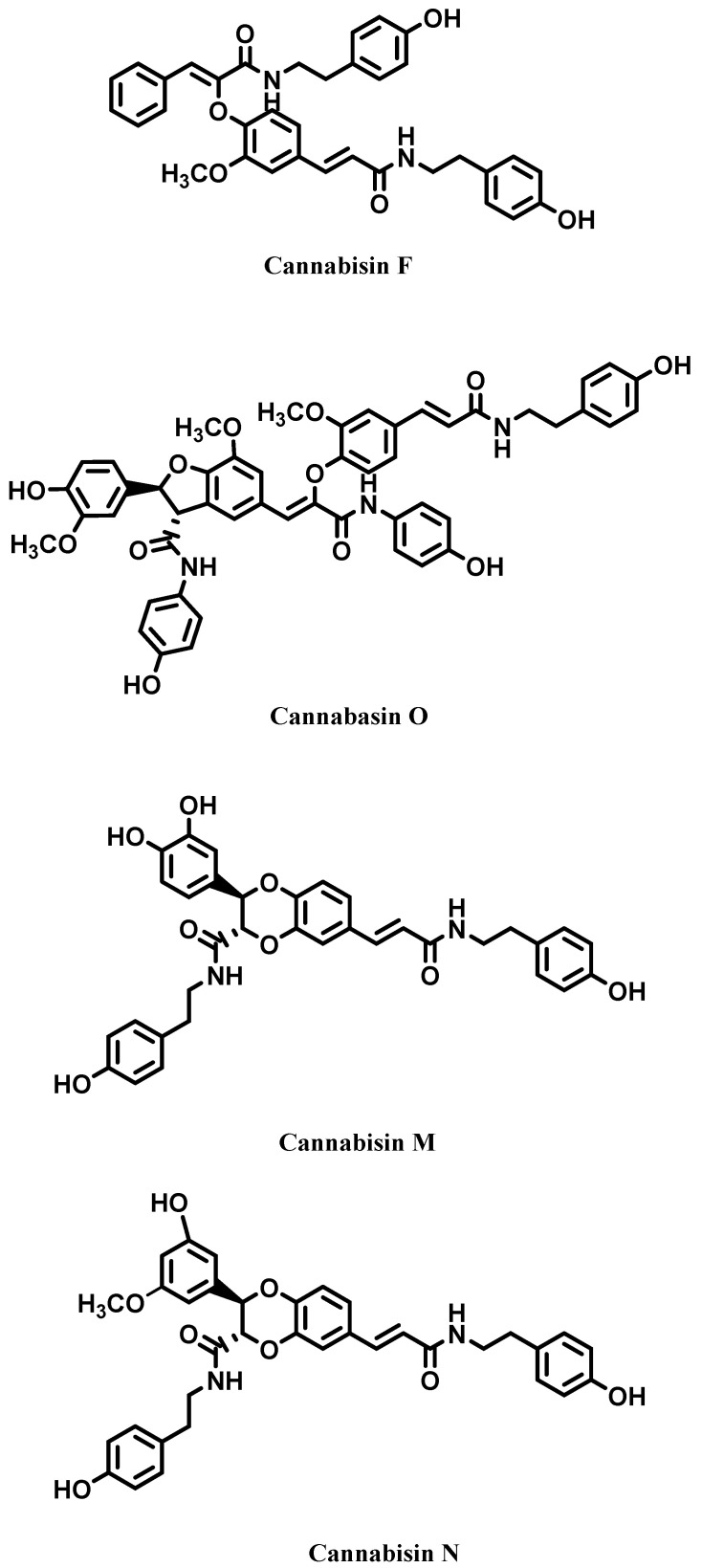
Cannabisins from *Cannabis sativa*.

**Figure 5 molecules-28-03387-f005:**
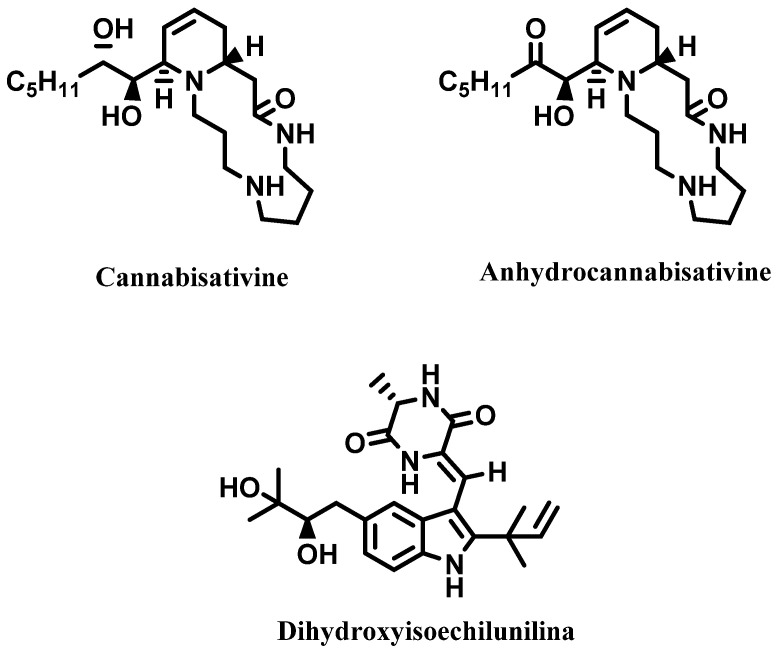
Alkaloids from *Cannabis sativa*.

**Figure 6 molecules-28-03387-f006:**
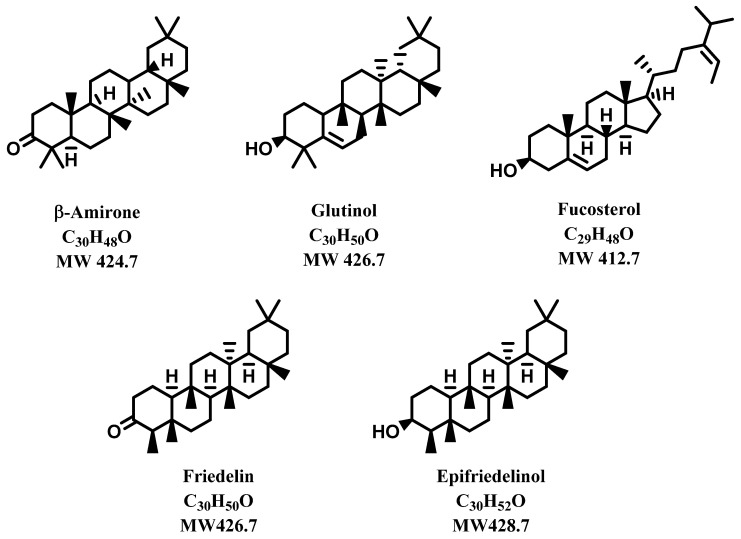
Triterpenoids from *Cannabis sativa*.

**Figure 7 molecules-28-03387-f007:**
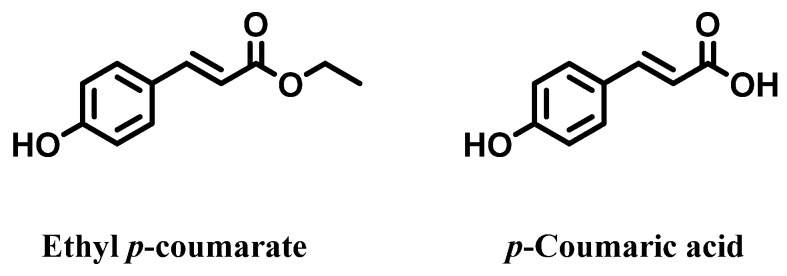
*p*-Coumaric acid and its ethyl ester in the roots of *Cannabis sativa*.

**Figure 8 molecules-28-03387-f008:**
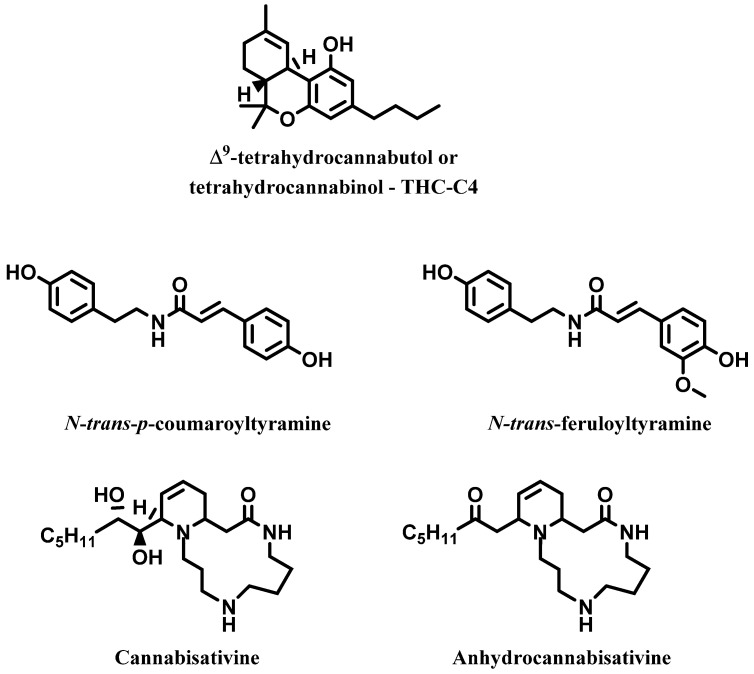
Δ^9^-tetrahydrocannabutol and four nitrogenous compounds isolated from roots of *C. sativa*.

**Figure 9 molecules-28-03387-f009:**
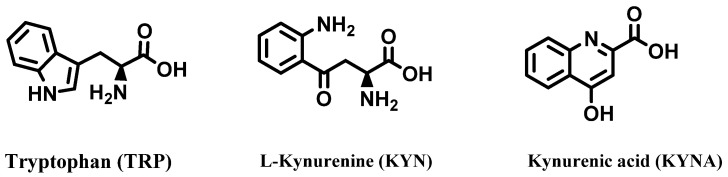
Tryptophan, kynurenine, and kynurenic acid from leaves, stems, and roots of *Cannabis sativa*.

**Table 1 molecules-28-03387-t001:** Selection of cannabinoids isolated from *Cannabis sativa* L.

Code	Name	Formula	MW ^a^
Δ^9^-THC	δ-9-Tetrahydrocannabinol	C_21_H_30_O_2_	314.55
Δ^8^-THC	δ-8-Tetrahydrocannabinol	C_21_H_30_O_2_	314.52
Δ^9^-THCP	(−)-*trans*-Tetrahydrocannabiphorol	C_23_H_34_O_2_	342.52
THCV	Tetrahydrocannabivarin	C_19_H_26_O_2_	286.45
Δ^9^-THCA-A	δ-9-Tetraidrocannabinolic acid A	C_22_H_30_O_4_	358.47
NA ^b^	Anhydrocannabimovone	C_21_H_28_O_3_	328.51
CBN	Cannabinol	C_21_H_26_O_2_	310.43
7-OH-CBD	7-Hydroxycannabinol	C_21_H_30_O_3_	330.46
CBNA	Cannabinolic acid	C_22_H_30_O_4_	358.47
CBD	Cannabidiol	C_21_H_30_O_2_	314.46
CBDD	Cannabidiol dimethyl ether	C_23_H_34_O_2_	342.52
CBD-C2	Cannabidiorcol	C_17_H_22_O_2_	258.35
CBD-DER1	1,2-Dihydroxycannabidiol	C_21_H_33_O_4_	348.36
CBD-DER2	3,4-Dehydro-1,2-dihydroxycannabidin	C_21_H_31_O_3_	330.37
CBD-DER3	Hexocannabitriol	C_21_H_31_0_4_	348.64
CBC	Cannabichromene	C_21_H_30_O_2_	314.46
CBG	Cannabigerol	C_21_H_32_O_2_	316.48
CNM	Cannabimovone	C_21_H_30_O_4_	346.46
CBND	Cannabinodol	C_21_H_26_O_2_	310.43
CBE	Cannabielsoin	C_21_H_30_O_4_	346.46
CBF	Cannabifuran	C_21_H_26_O_2_	310.24
CBL	Cannabicyclol	C_21_H_30_O_2_	314.46
CBT	Cannabitriol	C_21_H_30_O_3_	330.46
CBR	Cannabiripsol	C_21_H_32_O_4_	348.23
CBCA	Cannabichromenic acid	C_22_H_30_O_4_	358.21
CBGA	Cannabigerolic acid	C_22_H_32_O_4_	360.49
CBDA	Cannabidiolic acid	C_22_H_30_O_4_	358.47
CBDP	Cannabiphorol	C_23_H_34_O_2_	342.52
CBDD	Cannabitwinol	C_43_H_60_O_4_	64033
CBDV	Cannabidivarin (Cannabidivarol)	C_19_H_22_O_2_	282.38
CBTC	Cannabicitran	C_21_H_30_O_2_	314.46

^a^ Molecular weight in Daltons (DA); ^b^ NA = not available.

**Table 2 molecules-28-03387-t002:** Chemotype of *Cannabis sativa* L. depending on the THCA/CBDA ratio.

THCA/CBDA	>1	Chemotype I
	<1	Chemotype III
	>1–<1	Chemotype II
High CBGA		Chemotype IV
No cannabinoids		Chemotype V

**Table 3 molecules-28-03387-t003:** Selected mono and sesquiterpenes, together with some non-terpenoidic compounds and cannabinoids obtained by hydrodistillation of *Cannabis sativa* inflorescences.

**Monoterpenes**					
α-Thujene	Sabinene	α-Pinene	Camphene	α-Terpinene	β-Myrcene
β-Pinene	α-Phellandrene	1,8-Cineole	(*Z*)-β-Ocimene	(*E*)-β-Ocimene	*p*-Cymene
Limonene	Δ3-Carene	α-Terpinene	γ-Terpinene	Fenchone	Terpinolene
Terpinen-4-ol	Linalool	Fenchol	Camphor	Isoborneol	Borneol
Menthol	α-Terpineol	Citronellol	Pulegone	Geranyl acetate	*p*-Cymenene
Geraniol	Nerol	Geranial	Neral	2-Heptnone	Heptanal
2-Pinen-10-ol	*p*-Cymen-8-ol	*trans*-Pinocarveol	Myrtenol	Linalool acetate	Safranal
**Sesquiterpenes**					
α-Ylangene	α-Copaene	β-Elemene	β-Longipinene	Z-Caryophyllene	β-Caryophyllene
α-Longipinene	α-Humulene	β-Guaiene	β-Acoradiene	γ-Himachalene	α-Selinene
β-Himacalene	δ-Amorphene	γ-Cadinene	δ-Cadinene	α-Cadinene	*E*-Nerolidol
Germacrene B	Spathunelol	Viridiflorol	Ledol	Humulene epoxide	*epi*-α-Bisabolol
α-Cedrene	Cedrol	β-Eudesmol	α-Bisabolol	Valencene	α-Calacorene
β-Patchoulene	γ-Gurjunene	α-Curcumene	β-Selinene	γ-Gurjunene	Globulol
Sativene	α-Santalene	Sesquithujene	α-Zingiberene	β-Curcumene	Palustrol
Iso-Caryophyllene	α-Bulnesene	Aromadendrene	6,9-Guaiadiene	Iso-Valencenol	γ-Muurolene
Dehydro-aromadendrene	β-*cis*-Farnesene	Caryophyllene oxide	α-*cis*-Bergamotene	α-*trans*-Bergamotene	*Allo*-aromadendrene
**Not Terpenoidic Compounds**					
2-Heptanone	Heptanal	Hexyl hexanoate	Nonanal	Eugenol	Estragol
**Cannabinoids**					
Cannabidivarol	Cannabicitran	Cannabidiol	Cannabichromene	Cannabigerol	Δ-9-THC

**Table 4 molecules-28-03387-t004:** Neurological disfunctions for which cannabis-based treatments have been employed.

Code	Name	Formula
Epilepsy (Lennox–Gastaut and Dravet syndromes)	Cannabidiol (Epidiolex^®^)	Phase III, Regulatory approval
Chronic pain	THC, nabiximols	Phase II RCTs
Schizophrenia	CBD	Phase II
Sleep disturbance	THC, nabilone, nabiximols	Phase II–III
Tourette syndrome	THC, cannabis	Phase II, observational studies
Parkinson’s disease symptoms	THC, CBD, cannabis	Observational studies
Post-traumatic stress disorder	Cannabis	Observational studies
Dementia with agitation	THC, cannabis	Observational studies
Social anxiety	CBD	Phase II, observational studies

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
