# Peer review of "Recent Research on Cannabis sativa L.: Phytochemistry, New Matrices, Cultivation Techniques, and Recent Updates on Its Brain-Related Effects (2018–2023)"

_molecules, 2023, doi:10.3390/molecules28083387_

Round 1

Reviewer 1 Report

1.Introduction

Talking about the phytochemical profile of C.sativa, it would be helpful to deep the aspect of the element profile of such species, which recently has demonstrated to be useful not only for determining potential toxicity issues of C.sativa but also for traceability purposes.

Please, see recent literature such as:

-Nava, V.; Albergamo, A.; Bartolomeo, G.; Rando, R.; Litrenta, F.; Lo Vecchio, G.; Giorgianni, M.C.; Cicero, N. Monitoring Cannabinoids and the Safety of the Trace Element Profile of Light Cannabis sativa L. from Different Varieties and Geographical Origin. Toxics 202210, 758. https://doi.org/10.3390/toxics10120758

-Amendola, G.; Bocca, B.; Picardo, V.; Pelosi, P.; Battistini, B.; Ruggieri, F.; Barbini, D.A.; De Vita, D.; Madia, V.N.; Messore, A.; et al. Toxicological aspects of cannabinoid, pesticide and metal levels detected in light Cannabis inflorescences grown in Italy. Food Chem. Toxicol. 2021156, 112447

-Bengyella, L.; Kuddus, M.; Mukherjee, P.; Fonmboh, D.J.; Kaminski, J.E. Global impact of trace non-essential heavy metal contaminants in industrial cannabis bioeconomy. Toxin Rev. 202141, 1215–1225

Author Response

  1. = Referee; A. = Author

Referee #1

R.

1.Introduction 

Talking about the phytochemical profile of C. sativa, it would be helpful to deep the aspect of the element profile of such species, which recently has demonstrated to be useful not only for determining potential toxicity issues of C. sativa but also for traceability purposes.

A.

The introduction has been expanded according to the referee suggestion adding further references. 

Reviewer 2 Report

The topic of the review is interesting and useful for other research. Basically, it summarizes well the recent knowledge of the last years, so I consider it suitable for publication as a review.

A few comments, however, to improve the manuscript:

Abstract: I suggest writing THC in full at the first mention, with the abbreviation next to it.

Keywords: it is worth listing those not mentioned in the title and abstract. I suggest using other keywords instead of "Cannabis sativa L.", "novel matrices".

Introduction: 2nd paragraph: Urticales (written in one)

Cannabis sativa L. full name should only be written out in full first, then C. sativa is sufficient (need be corrected throughout the article)

Table 1: legend of „MW” abbreviation, unit of measurement missing. source(s) for this table? Likewise, for the other figures/tables, references are missing (in the title or in a note, according to the journal's specifications)

4th page, 2nd paragraph: area data are only shown for Italy. I suggest that either detail the global trend, or several countries, or delete the data for Italy.

Page 4, 2nd paragraph from the bottom: there are two end-of-sentence bullet points at the end - one to be deleted.

Molecular structure diagrams should be uniformly the same size for molecular units (e.g. Figure 4 is larger than Figure 5), and a uniform font/letter size for molecular names is also recommended.

Page 11: subsection 2.2: the 3 oils listed are not all seed oils, please clarify, re-reading the original publication cited.

Page 13: The title of subsection 3.2 is incorrect. Perhaps it would be better: soilless growing technologies. This chapter should indeed list new and concrete results. These technologies existed before, and there are several much older publications on them. After a short introduction, it should be indicated here what is really new in the period under review.

Page 14, chapter 3.4 it is perhaps more appropriate to talk about "symbiotes", as mentioned in the title (microbiota- is too wide term)

Author Response

  1. = Referee; A. = Author

Referee #2

The topic of the review is interesting and useful for other research. Basically, it summarizes well the recent knowledge of the last years, so I consider it suitable for publication as a review.

A few comments, however, to improve the manuscript:

R.

Abstract: I suggest writing THC in full at the first mention, with the abbreviation next to it.

A.

Done

R.

Keywords: it is worth listing those not mentioned in the title and abstract. I suggest using other keywords instead of "Cannabis sativa L.", "novel matrices".

A.

Done

Introduction: 2nd paragraph: Urticales (written in one)

A.

Done

R.

Cannabis sativa L. full name should only be written out in full first, then C. sativa is sufficient (need be corrected throughout the article)

A.

Done

R.

Table 1: legend of „MW” abbreviation, unit of measurement missing. source(s) for this table? Likewise, for the other figures/tables, references are missing (in the title or in a note, according to the journal's specifications)

A.

The legend of MW has been enclosed; adding references to the reported data would be prohibitive and even usesless.

R.

4th page, 2nd paragraph: area data are only shown for Italy. I suggest that either detail the global trend, or several countries, or delete the data for Italy.

A.

The cited paragraph has been removed as suggested.

R.

Page 4, 2nd paragraph from the bottom: there are two end-of-sentence bullet points at the end - one to be deleted.

A.

Done

R.

Molecular structure diagrams should be uniformly the same size for molecular units (e.g. Figure 4 is larger than Figure 5), and a uniform font/letter size for molecular names is also recommended.

A.

All molecular structures reported in the figures have been revised and size as well as font/letter size have been standardized.

R.

Page 11: subsection 2.2: the 3 oils listed are not all seed oils, please clarify, re-reading the original publication cited.

A.

According to the referee’s suggestion the statement has been corrected

R.

Page 13: The title of subsection 3.2 is incorrect. Perhaps it would be better: soilless growing technologies. This chapter should indeed list new and concrete results. These technologies existed before, and there are several much older publications on them. After a short introduction, it should be indicated here what is really new in the period under review.

A.

The reviewer is indeed right. As soilless and generally controlled cultivation are not new, the old paragraph title has been now replaced with Soil less growing technologies applied to cannabis: hydroponic, aquaponic, and aeroponic cultivation – recent updates and an opportune sentence has been added in the text.

R.

Page 14, chapter 3.4 it is perhaps more appropriate to talk about "symbiotes", as mentioned in the title (microbiota- is too wide term)

A.

Thank you. The title of paragraph 3.4 has now been changed to Cannabis symbiotic microorganisms

Reviewer 3 Report

In the current manuscript, Siracusa et al. reviewed the phytochemistry, new matrices, new agronomic techniques and new biological activities developed in the five last years in cannabis. Although the topic is attractive, there are some concerns that should be addressed. 

-Generally, the manuscript has some typographical and grammatical errors. 

-The paper title is well stated, and it is informative and concise. 

-Abstract is not well structured and should be re-written.  

-The introduction was not well written, and it is too briefly presenting the subject and research problem. 

The line numbers have been omitted from the manuscript, which makes it difficult to state where revisions should be made. 

Paragraph 1 of the introduction: The origin of this species! which species??? 

It is better, first, to introduce different applications of cannabis as well as hemp and drug-type cannabis. So, I suggest the following sentences: 

"Cannabis has been widely cultivated due to its industrial (DOI: 10.3906/bot-1907-15), ornamental (10.3390/plants11182383), nutritional (10.3390/plants11233330), medicinal, and recreational (10.1016/j.biotechadv.2022.108074) potentials. From regulatory and application perspectives, cannabis plants are categorized based on the level of Δ9-tetrahydrocannabinol (THC), one of the most important phytocannabinoids (10.1146/annurev-arplant-081519-040203). Plants are generally classified and regulated as industrial hemp if it contains less than 0.3 % THC in the dried flower (this level varies by country) or drug-type with more than this threshold (10.1016/j.indcrop.2020.113026)." 

The main objectives of this review manuscript should be mentioned in the introduction section. 

-Subsections (1-1 to 1-5) should be presented as the subsection of secondary metabolites of cannabis NOT INTRODUCTION. 

The authors have missed providing citations to support several sentences (For instance section 3). 

For example: Section 3-1: Asexual reproduction of Cannabis requires only a single plant, which allows for the multiplication of a single genotype for the commercial production of individuals owning desirable pharmaceutical traits. Please provide appropriate citations. 

Section 3-1: "This high variability of ... the use of cannabis for medical purposes". Please provide appropriate citations. 

Section 3-3: "plant morphology, metabolism, ... and duration of light”. Please provide appropriate citations. 

Page 16: The authors did not present Table 4. 

- The conclusion should be improved. I suggest that the authors mention the limitations of the current advances of cannabis in the conclusion part and specify the future outlooks. 

The line numbers have been omitted from the manuscript, which makes it difficult to state where revisions should be made. Please put line numbers for the further round of review. 

Author Response

  1. = Referee; A. = Author

Referee #3

In the current manuscript, Siracusa et al. reviewed the phytochemistry, new matrices, new agronomic techniques and new biological activities developed in the five last years in cannabis. Although the topic is attractive, there are some concerns that should be addressed. 

R.

Generally, the manuscript has some typographical and grammatical errors.

A.

A general correction has been carried out hoping to have corrected everything.

  1.  

Abstract is not well structured and should be re-written.

A.

The abstract has been partially modified.

R.

The introduction was not well written, and it is too briefly presenting the subject and research problem.

A.

As replied to the first referee the introduction has been modified and expanded.

  1.  

The line numbers have been omitted from the manuscript, which makes it difficult to state where revisions should be made.

A.

The template used by the journal does not allow entry the line numbers.

  1.  

Paragraph 1 of the introduction: The origin of this species! which species.

  1.  

The correct name of the species has been enclosed.

R.

It is better, first, to introduce different applications of cannabis as well as hemp and drug-type cannabis. So, I suggest the following sentences: 

 "Cannabis has been widely cultivated due to its industrial (DOI: 10.3906/bot-1907-15), ornamental (10.3390/plants11182383), nutritional (10.3390/plants11233330), medicinal, and recreational (10.1016/j.biotechadv.2022.108074) potentials. From regulatory and application perspectives, cannabis plants are categorized based on the level of Δ9-tetrahydrocannabinol (THC), one of the most important phytocannabinoids (10.1146/annurev-arplant-081519-040203). Plants are generally classified and regulated as industrial hemp if it contains less than 0.3 % THC in the dried flower (this level varies by country) or drug-type with more than this threshold (10.1016/j.indcrop.2020.113026)." 

A.

The referee’s suggestion has been accepted and the proposed sentence has been enclosed.

R.

The main objectives of this review manuscript should be mentioned in the introduction section. 

A.

According to the referee’s suggestion an opportune statement has been added.

R.

Subsections (1-1 to 1-5) should be presented as the subsection of secondary metabolites of cannabis.

A.

According to the referee’s comment a new section entitled 2. Secondary metabolites of cannabis has been added  

R.

The authors have missed providing citations to support several sentences (For instance section 3). 

A.

Appropriate citations have now been added accordingly.

R.

For example: Section 3-1: Asexual reproduction of Cannabis requires only a single plant, which allows for the multiplication of a single genotype for the commercial production of individuals owning desirable pharmaceutical traits. Please provide appropriate citations. 

A.

Appropriate citations have now been added accordingly.

R.

Section 3-1: "This high variability of ... the use of cannabis for medical purposes". Please provide appropriate citations. 

A.

Appropriate citations have now been added accordingly.

R.

Section 3-3: "plant morphology, metabolism, ... and duration of light”. Please provide appropriate citations. 

A.

Appropriate citations have now been added accordingly.

R.

Page 16: The authors did not present Table 4. 

A.

The table has been now inserted

R.

The conclusion should be improved. I suggest that the authors mention the limitations of the current advances of cannabis in the conclusion part and specify the future outlooks. 

A.

The conclusion have been totally rewritten accepting also the referee’s suggestions

R.

The line numbers have been omitted from the manuscript, which makes it difficult to state where revisions should be made. Please put line numbers for the further round of review. 

A.

See previous comment

Round 2

Reviewer 3 Report

All the comments have been addressed. I think that the current version of the manuscript can be published in the journal.

Author Response

Dear Editor,

According to the editor’s suggestion, we have modified the conclusion section enclosing all subjects discussed in the MS, namely compounds, cultivation and the recent studies on the brain effects of cannabis products, underlying, as mentioned in the MS, the risks associated to the toxicity of some components and the beneficial effects associated to other ones.

Furthermore, the new revised version includes all previous changes, which have been accepted by referees, while in the new revised version, here enclosed, since the changes only concern the conclusion section, the changes have been highlighted in yellow.

Hoping to have been quite clear, and the changes meet with your approval, please accept my best regards.

Sincerely

Giuseppe Ruberto
